# Nutrition and microRNAs: Novel Insights to Fight Sarcopenia

**DOI:** 10.3390/antiox9100951

**Published:** 2020-10-02

**Authors:** Alessandra Barbiera, Laura Pelosi, Gigliola Sica, Bianca Maria Scicchitano

**Affiliations:** 1Sezione di Istologia ed Embriologia, Dipartimento di Scienze della Vita e Sanità Pubblica, Fondazione Policlinico Universitario A. Gemelli IRCCS, 00168 Roma, Italy; Alessandra.Barbiera@unicatt.it (A.B.); Gigliola.Sica@unicatt.it (G.S.); 2DAHFMO-Unità di Istologia ed Embriologia Medica, Sapienza Università di Roma, 00161 Roma, Italy; laura.pelosi@uniroma1.it

**Keywords:** ageing, autophagy, fructose, hormesis, inflammation, nutrition, oxidative stress, skeletal muscle, TNF, uric acid, vitagene

## Abstract

Sarcopenia is a progressive age-related loss of skeletal muscle mass and strength, which may result in increased physical frailty and a higher risk of adverse events. Low-grade systemic inflammation, loss of muscle protein homeostasis, mitochondrial dysfunction, and reduced number and function of satellite cells seem to be the key points for the induction of muscle wasting, contributing to the pathophysiological mechanisms of sarcopenia. While a range of genetic, hormonal, and environmental factors has been reported to contribute to the onset of sarcopenia, dietary interventions targeting protein or antioxidant intake may have a positive effect in increasing muscle mass and strength, regulating protein homeostasis, oxidative reaction, and cell autophagy, thus providing a cellular lifespan extension. MicroRNAs (miRNAs) are endogenous small non-coding RNAs, which control gene expression in different tissues. In skeletal muscle, a range of miRNAs, named myomiRNAs, are involved in many physiological processes, such as growth, development, and maintenance of muscle mass and function. This review aims to present and to discuss some of the most relevant molecular mechanisms related to the pathophysiological effect of sarcopenia. Besides, we explored the role of nutrition as a possible way to counteract the loss of muscle mass and function associated with ageing, with special attention paid to nutrient-dependent miRNAs regulation. This review will provide important information to better understand sarcopenia and, thus, to facilitate research and therapeutic strategies to counteract the pathophysiological effect of ageing.

## 1. Introduction

It is generally accepted that the progressive age-related reduction in skeletal muscle mass and strength, a condition known as sarcopenia [1], is implicated in an increased incidence of falls, disability, and loss of independence [2,3,4]. Moreover, decreased muscle strength is also highly predictive of adverse outcomes and may cause mortality in older persons [5]. The mechanisms that underlie sarcopenia are not yet completely elucidated, but it is likely that sarcopenia is the result of multifactorial events, such as a reduction in number and activity of satellite cells [6], mitochondrial dysfunction [7,8], elevated level of inflammation [9], increased ROS production [10] and imbalance between protein synthesis and breakdown [11,12,13,14] (Figure 1). Indeed, in the elderly, the proteolytic processes are not accompanied by an adequate protein synthesis within the physiological turnover, and muscle cells lose progressively the sensitivity to anabolic stimuli, thus manifesting the so-called “anabolic resistance” [15,16]. Protein balance is regulated by different factors, each susceptible to alterations during ageing; among them are changes in hormone levels [17,18], a decreased physical activity, and inadequate food intake [19,20]. Food intake falls by around 25% between 40 and 70 years of age [21], and there is growing evidence that correlates poor nutrition and adverse effects on muscle in the elderly, suggesting that the maintenance of adequate nutritional intake could be an effective strategy for preventing or treating sarcopenia [22].

MicroRNAs (miRNAs) are endogenous small non-coding RNAs, containing approximately 22 nucleotides, which control gene expression by targeting mRNAs and triggering either the translation repression or RNA degradation [23,24]. MiRNAs are required for many biological processes, such as intercellular communication, differentiation, and proliferation [25,26]. In skeletal muscle, a range of miRNAs, named myomiRNAs, has been identified, and includes miRNA-1, miRNA-133a, miRNA-133b, miRNA-206, miRNA-208b, miRNA-486, miRNA-499 [27,28,29]. MyomiRNAs regulate multiple aspects of skeletal muscle, since they are involved in many physiological processes, such as growth, development, and maintenance of muscle mass and function [30,31,32]. Consequently, alterations of miRNAs expression may occur during ageing, and can be associated with pathological conditions [30,33,34,35,36].

This review aims to present and to discuss some of the most relevant molecular mechanisms related to the pathophysiological effect of sarcopenia. Besides, we explored the role of nutrition as a possible way to counteract the loss of muscle mass and function associated with ageing, lading a special focus on nutrient-dependent miRNAs regulation, which represents an important component to fight sarcopenia. This review will provide essential information in a general attempt to better understand sarcopenia and, thus, facilitate research and therapeutic strategies in the future.

## 2. Nutrition-Dependent microRNA Regulation of Skeletal Muscle Regeneration

Although adult skeletal muscle is composed of fully differentiated fibers, it retains the capacity to regenerate in response to injury. Muscle regeneration is a highly coordinated process that leads to a morpho-functional recovery of injured tissue through the activation and differentiation of muscle stem cells, maturation of newly formed muscle fibers, and remodeling of extracellular matrix [37,38]. The decrease of skeletal muscle regenerative capacity has been observed in both human and mice sarcopenic muscle, and it seems to be the primary consequence of satellite cells ageing [39,40]. The severe alteration in the functionality of satellite cells in senescent muscle can be caused by either extrinsic factors or intrinsic events, including defects in self-renewing mechanisms, exhaustion by forced differentiation, as well as apoptosis and alteration of muscle environment [41,42].

An elegant study by Conboy et al. demonstrated the rejuvenation of aged progenitor cells by exposure to a young systemic environment, supporting the heterochronic transplantation experiments, in which satellite cells, in aged mice that had been paired with young mice, showed marked improvements in functionality [43]. Similarly, specific nutrients may also promote a more rejuvenating systemic milieu enhancing satellite cell function and favoring healthy aging both in in vivo and in vitro experimental models [44]. Likewise, satellite cells in young mice that had been paired with old mice showed a decline in functionality [43,45,46]. These data suggest that there is a strong contribution of the environment to the satellite cell ageing phenotype, including the dysregulation of signals from either the myofibers or the circulatory system. Nevertheless, additional experimental evidence revealed that ageing induces intrinsic alterations in muscle stem-cell regenerative functions, which cannot be rejuvenated by a young host environment [39]. This is due to the modulation of the transcriptional and epigenetic network that regulates distinct fates of stem-cell progeny during ageing.

Among other factors, miRNAs play an important role in the modulation of stem cell function and activity, muscle homeostasis, and have been involved in different neuromuscular diseases. In particular, Chuang et al. elegantly demonstrated that the ablation of miRNAs in satellite cells leads to a reduced number of these cells, mild atrophy with ageing, and an impaired regenerative capability of muscle fibers upon injury [47]. Recently, several studies demonstrated that miR-1, miR-206, miR-133, miR-188, and miR-27 are potential regulators of the muscle regeneration process. In particular, miR-1, miR-133, and miR-206 are induced upon satellite cell commitment and differentiation, and their increased expression promotes the differentiation of these cells [48,49]. Besides, the local injection of a cocktail of miRNAs, including miR-1, miR-133, and miR-206, in rat skeletal muscle injury model, enhanced regeneration, and prevented fibrosis [50]. Moreover, in regenerating muscle, miR-27 plays a crucial role in downregulating Pax3 expression in order to stimulate myogenesis, while its inhibition in injured muscle delays muscle regeneration [51].

It has been well established that several nutrients such as amino acids and glucose may alter the expression of miRNAs [12,52,53,54,55]. Leucine has been shown to induce the proliferation of satellite cells and to increase the size and strength of regenerating fibers [56,57]. Moreover, Drummond et al. (2009) showed that acute essential amino acids (EAAs) ingestion elicited robust increases in miR-1, miR-23a, miR-208b, and miR-499 expression, with an accompanying increase in MyoD1 and Follistatin Like 1 mRNA expression, and a decrease in myostatin and MEF2C mRNA expression in human skeletal muscle [52]. It has been also reported by Iannone et al. [58] that miR-133a/b and miR-206 appear to be directly or indirectly regulated by the mammalian target of rapamycin (mTOR) [59], the main mediator of cellular nutrient sensing, and a key regulator of skeletal muscle growth and hypertrophy [60]. According to these studies, Zhang and al. [36] proposed a model for nutrient—mTOR-myomiR signaling, where mTOR may affect the expression of miR-133a/b and miR-206, through the regulation of MyoD transcription factor. In agreement with this model, under low nutrient conditions such as amino acid and glucose starvation, mTOR is inactive and, consequently, unable to induce MyoD resulting in the downregulation of miR-133a/b and miR206 (Figure 2).

Although further experiments are needed to elucidate the molecular mechanisms that regulate the effect of specific nutritional compounds on miRNAs expression in skeletal muscle regeneration, these findings demonstrate a significant miRNAs response to essential amino acid supplementation, and suggest a key role for these molecules in regulating the homeostasis of muscle tissue.

## 3. Nutrition-Dependent microRNA Regulation of Inflammageing

Ageing is associated with a chronic low-grade inflammatory state known as “inflammageing”, characterized by a 2- to 3-fold elevation in circulating inflammatory mediators [61]. Pro-inflammatory cytokines, such as TNF, IL-6, and C-reactive protein (CRP), are key components in this chronic inflammatory condition. Recently, experimental evidence demonstrates that the above mentioned pro-inflammatory cytokines significantly increase ageing in skeletal muscle cells, and play a key role in the complex network that connects inflammatory signals with ageing-related disability and mortality [62,63,64].

In particular, elevated serum levels of IL-6 and TNF are markers of functional frailty and predictors of poor prognosis in the elderly [65], and increased levels of cellular IL-6 production are a significant predictor of sarcopenia [66]. Besides, elevated levels of CRP predict mortality and functional decline in older subjects [67]. Notably, the chronic inflammatory ageing process depends not only on the increased concentration of pro-inflammatory cytokines, but also on a reduction in the levels of anti-inflammatory cytokines [68].

Although the molecular signaling involved in the interaction between inflammageing and muscle loss is not yet completely understood, recent in vivo studies demonstrate that the increased low-grade inflammation may result in the activation of catabolic pathways favoring protein breakdown and inhibiting protein synthesis, ultimately leading to age-related muscle wasting [69,70,71,72].

In muscle, pro-inflammatory cytokines such as TNF regulate sarcopenia through the activation of the nuclear factor kappa B (NF-κB) transcription factor which, in turn, may activate the ubiquitin-proteasome system [73]. NF-kB is maintained in the inactive state by the binding with a family of inhibitory proteins called IκB. The increase in the TNF level induces activation of an IκB kinase (IKK) complex that phosphorylates IκB, which, in turn, leads to its degradation mediated by the proteasome system. This degradation of IκB allows for NF-κB to translocate to the nucleus and to activate the transcription of several κB-dependent genes [74]. In particular, under conditions of chronic inflammation, high levels of NF-κB expression activate the ubiquitin-proteasome system which involves an enzymatic cascade that begins with the ubiquitination of protein substrates and terminates with the hydrolysis of targeted protein to small peptides or amino acids, resulting in protein degradation and muscle wasting [71,75].

As mentioned previously, specific miRNAs, named myomiRNAs, are known to be associated with the skeletal muscle [27,29,76], where they play a crucial role, by targeting genes involved in different processes such as development, differentiation, and regeneration [77]. Experimental evidence demonstrated that the expression of these myomiRNAs can be dysregulated during ageing and contribute to the resistance of older muscles to anabolic stimuli [78]. It has been reported that the cytokine named TNF-weak-inducer of apoptosis can induce muscle wasting through the regulation of miRNAs including miRNA-1, miRNA-133a, and miRNA133b, involved in the growth of mouse skeletal muscle [79]. Besides, a down-regulation of miRNA-133b and miRNA-206 was observed by Georgantas et al. in the muscle of patients with inflammatory myopathy [80], and the expression of these miRNAs has been correlated by Iannone et al. with the nutritional status, revealing a mediating effect of nutrition on the relationship between sarcopenia and myomiRNAs [58].

In addition to the myomiRNAs, other miRNAs are critical regulators for both pro-inflammatory cytokines and skeletal muscle function [81,82]. For instance, miRNA-155 significantly increases upon muscle injury and in mdx mice, the mouse model of Duchenne muscular dystrophy. By a genetic approach, M. Nie et al. demonstrated that the ablation of miRNA-155 expression severely compromised skeletal muscle regeneration, largely owing to aberrant macrophage activation and disrupted balance between the expression of pro- and anti-inflammatory cytokines [83].

A recent RNA sequencing study performed by Mercken et al. (2013) revealed the differential expression of miRNAs in the skeletal muscles of old and young rhesus monkeys [84]. Besides, Xie et al. (2013) observed that the expression of miR-181a was downregulated in old muscle and its reduction resulted in an increased expression of the cytokines TNF, IL-6, IL-1b in skeletal muscle during the ageing process [85].

It has been well established that malnutrition and sarcopenia are closely correlated with inflammation [9]. Nutrients such as glucose and amino acids can modulate the expression of miRNAs [52,53,54,55], and caloric restriction can revert the level of miR-181a, suggesting a significant role of nutrition in the modulation of the inflammatory pathway.

Importantly, significant positive or negative correlations were found between miR-133b and miR-206 levels and albumin and ferritin, respectively [80], where decreased albumin and elevated ferritin levels are characteristic features of inflammation, besides being markers of nutritional status [86,87,88].

Although so far there is only a restricted number of studies regarding the molecular mechanisms involved in the interconnection between miRNAs, nutrition, and sarcopenia, it is reasonable to assume that dietary interventions may represent an efficient strategy to help prevent or counteract the loss of muscle mass and functionality that occurs in ageing.

## 4. Nutrition-Dependent microRNA Regulation of Mitochondrial Dysfunction

### 4.1. Autophagy

Autophagy is a highly evolutionarily conserved catabolic process through which misfolded proteins and dysfunctional organelles are degraded and recycled by autophagosomes that are then delivered to the lysosomal machinery to prevent waste accumulation [89,90]. While the basal level of autophagy is essential for the physiological turnover of old or damaged organelles, the dysregulation of autophagy signaling may cause cellular stress and death as a result of cellular atrophy or, alternatively, of apoptotic program induction [91].

Several findings indicate that autophagy becomes progressively dysfunctional during ageing, and this effect seems to be related to the accumulation of damaged cellular components such as defective mitochondria, which in turn may induce increased levels of reactive oxygen species (ROS) and trigger apoptotic events [92,93]. In particular, it has been demonstrated that in aged muscles, both excessive and defective autophagy may result in the onset of sarcopenia [94]. Indeed, the deficiency of basal autophagy can result in the abnormal aggregation of misfolded proteins, while excessive autophagy can also cause cellular stress and induce the loss of skeletal muscle mass due to increased protein degradation [95].

One of the most important proteins involved in the regulation of skeletal muscle autophagy is mTOR, a highly conserved serine/threonine kinase required for numerous aspects of cellular homeostasis [96]. MTOR phosphorylates several transcription factors involved in the autophagy process, thereby preventing their translocation to the nucleus [97]. An example is represented by the helix-loop-helix transcription factor TFEB, a member of the MITF (microphthalmia-associated transcription factor) family [98,99,100,101,102,103], that has been demonstrated to have a role in all the stages of autophagy process, from lysosomal biogenesis to autophagosome formation [102]. Under nutrition-rich conditions, mTOR phosphorylates TFEB that consequently is retained in the cytosol and is unable to stimulate autophagy gene expression [98,100,101]. Conversely, in response to nutrient deprivation, TEFB translocates to the nucleus to activate transcriptional targets leading to autophagy stimulation [104]. As reported by Lapierre LR et al., there is a TFEB homolog in C. elegans, named HLH-30, that plays a role similar to TFEB in the modulation of autophagy process [105]. HLH-30 translocates to the nucleus as a response of mTOR inhibition or nutrient deprivation, and it can regulate several genes involved in the autophagy process, supporting the concept that increased autophagic flux is likely critical to ensure a long lifespan [105,106]. Since mTOR-dependent regulation of TFEB activity is an evolutionarily conserved mechanism of the autophagic flux, there is an attempt to speculate that this process could provide a vital source of metabolites during periods of nutrient deprivation.

Another family of transcriptional factors involved in the regulation of the autophagy process, with a conserved role in ageing, is the Forkhead transcription factors (FoxO), which play a crucial role in the activation of the ubiquitin-proteasome system, but they are also involved in the activation of the autophagic/lysosomal pathway [107]. In particular, it has been demonstrated that several nutrient-signaling pathways can modulate FoxO activity [108]. Indeed, reduced INS-IGF1 signaling activates FoxO-dependent expression of genes involved in autophagy and proteostasis in several species [109,110] and extends longevity [108]. In 2015, Brown et al. demonstrated that FoxO3 might be post-transcriptionally regulated by miR-182, with a consequent modulation of genes involved in the autophagy/lysosome system. Moreover, they showed a critical role for miR-182 in the control of fuel usage and glucose homeostasis in skeletal muscle [111].

Among miRNAs involved in the regulation of the autophagy process in different species, miR-34 is up-regulated during ageing and may contribute to ageing process, by directly modulating the expression of autophagy-related proteins [112,113,114]. Recently, Yan Li et al. reported that miR-378 promotes autophagy through targeting PDK1, which is crucial in the activation of the PI3K/Akt signaling, but it also inhibits mitochondria-mediated intrinsic apoptosis by targeting Caspase 9. Since miR-378 is highly expressed in skeletal muscle, it is possible to speculate that failure to maintain the high levels of miR-378 in skeletal muscle would lead to increased vulnerability to cell death observed in muscle dystrophy or in sarcopenia. Notably, under metabolic stress conditions such as nutrient deprivation, miR-378 dramatically increases, suggesting its significant role in the cellular adaptation to dwindling nutrient resources [115].

### 4.2. ROS Imbalance

Mitochondria are important cellular organelles, with key regulatory functions in energy production, reactive oxygen species (ROS) balance, and in the control of cell death [116,117]. Mitochondrial function may be affected by cumulative damage to mitochondrial DNA, which occurs during ageing. The damaged mitochondrial DNA leads to an impairment of key electron transport enzymes and subsequent ROS generation, thus causing a decrease in energy production [118]. Although adequate levels of ROS play an important role in the maintenance of tissue homeostasis [119], age-related ROS overproduction has been proposed as one of the major contributors of the skeletal muscle decline that occurs with ageing [120,121]. Indeed, it not only generates oxidative damage of muscle, but it is also involved in regulating intracellular signal transduction pathways that play, directly or indirectly, a role in the impairment of skeletal muscle strength and functionality [62,72,122,123,124]. The opposite effects exerted by different concentrations of ROS can be justified considering the concept of hormesis, which is a process characterized by a biphasic response to environmental agent with a low-dose stimulation and high-dose inhibition [125]. Thus, skeletal muscle benefits from low doses of free radicals, whereas excessive free radicals concentration can impair its functions. Hence, efficient mechanisms of antioxidant defense have to be developed, especially in those tissues like skeletal muscle highly exposed to the oxidation process.

Antioxidants are present in different forms; some of them include enzymes such as superoxide dismutase (SOD), catalase (CAT), and glutathione peroxidase (GSH-Px), which converts free radicals into nontoxic forms, and others, represented by vitamins, carotenoids, and polyphenols, are introduced by the diet [10,126,127].

Vitamin C is a water-soluble antioxidant introduced in humans by dietary intake. Elevated levels of vitamin C are associated with a lower risk of hypertension, heart disease, and stroke [128]. This vitamin also promotes the regeneration of fat-soluble vitamin E in the cell membrane [129]. A protective effect of vitamin C supplementation against exercise-induced muscle damage was demonstrated by Jakeman and Maxwell. They also reported that vitamin E exerts antioxidant properties by scavenging ROS and boosting cellular anti-oxidative capacity to reduce oxidative damage [130]. Similarly, vitamin C and E supplementation has been shown to reduce muscle damage by Shafat and colleagues [131]. In a mouse model of muscle atrophy, the MLC/SOD1 G93A mice, characterized by progressive muscle atrophy associated with a significant reduction of muscle strength, alteration in the contractile apparatus, and mitochondrial dysfunction, the treatment with a derivate of vitamin E significantly reduced the toxic effect of ROS, partially rescuing muscle phenotype and muscle performance [132]. Moreover, a mixture of antioxidants, including vitamin E, vitamin A, zinc, and selenium has been shown to increase the anabolic response of all the muscles to leucine and the leucine-induced inhibition of protein degradation in rats [133].

Recently, particular attention has been paid to the polyphenols. These molecules, which are produced as secondary metabolites by the plants for protection against bacteria, fungi, and insects, display remarkable antioxidant properties [134]. Experimental studies performed in animal models showed that the dietary administration of polyphenols, such as resveratrol, in combination with treadmill exercise, exert beneficial effects which improve mitochondrial function, and reduce age-related decline in physical performance [135]. Similarly, the supplementation of another polyphenol represented by curcumin ameliorates exercise performance in rats [136]. Moreover, the Geny group demonstrated that intake of polyphenols starting at a young age restored muscle maximal mitochondrial oxidative capacity, normalized production of ROS, and enhanced antioxidant defense, therefore protecting aged muscle [137].

However, controversial data have been published regarding the relationship between antioxidant supplementation and muscle performance. In fact, human trials did not confirm the positive results obtained in animals. It has been shown that undesirable effects, such as the disruption of the endogenous antioxidant levels, may result from prolonged antioxidant supplementation, thus failing to counteract exercise-induced oxidative stress, and interfering with muscle adaptation to exercise [138,139,140]. Moreover, the long-term administration of vitamin C has been observed to prevent mitochondrial biogenesis, decreasing the expression of endogenous antioxidant enzymes [141].

Several reasons can be responsible for these contradictory results. In particular, it has been demonstrated that ROS are required for cellular adaptation to exercise and for the insulin-sensitizing capabilities of physical exercise in healthy humans. Besides, the health-promoting effects of physical exercise are abrogated by antioxidants such as vitamin C and E, and polyphenols. A potentially health-promoting process may be derived from transiently increased levels of oxidative stress, whereas an uncontrolled accumulation of oxidative stress may have pathological implications.

Recent studies revealed that the direct antioxidant properties of polyphenols are not the major mechanism of their action [142,143]. In fact, there is a poor bioavailability and very low concentrations of active polyphenols in target tissues. It seems likely that the antioxidant effects of polyphenols are mediated via the activation of various transcription factors, signaling pathways, and vitagenes. Vitagenes encode components of the heat shock protein (HSP), thioredoxin, and sirtuin protein systems, that show antioxidant and antiapoptotic activities [144,145,146,147,148]. In particular, the effects of polyphenols in the vitagene network can be demonstrated using silymarin (SM), a plant extract containing polyphenols. In fact, as reported by Surai et al., SM was shown to improve antioxidant defenses by upregulating heme oxygenase-1 (HO-1). In addition, SM consumption has been shown to be associated with decreased HSP70 expression in stressed cells, which indicates an improvement in anti-oxidant defenses. Finally, SM-related activation or the prevention of inhibition of sirtuins in stress conditions might be an essential adaptive mechanism responsible for maintaining the redox-regulated homeostasis in the cell and the whole body [149]. SIRT1 has been identified as a link between caloric restriction and longevity, and its overexpression is linked to increased lifespans for several organism models [150]. SIRT1 activation inhibits NF-κB signaling and increases oxidative metabolism, favoring the resolution of inflammation. SIRT1 exerts this effect directly by deacetylating the p65 subunit of NF-κB complex. SIRT1 activates AMPK, PPARα, and PGC-1α stimulating oxidative energy production; these factors inhibit NF-κB signaling and suppress inflammation. On the other hand, the expression of miR-34a, IFNγ, and ROS, induced by NF-κB signaling, down-regulates SIRT1 activity. The inhibition of SIRT1 disrupts oxidative energy metabolism and stimulates the NF-κB-induced inflammatory responses present in many chronic metabolic and age-related diseases [151].

Several miRNAs play a crucial role in the regulation of mitochondrial gene expression. For example, miR-1, a microRNA specifically induced during myogenesis, efficiently enters the mitochondria, where it stimulates the translation of specific mitochondrial genome-encoded transcripts. Moreover, miR-696 negatively affects fatty acid oxidation and mitochondrial function by targeting the transcription factor peroxisome proliferator-activated receptor γ coactivator 1α (PGC-1α), a master regulator of mitochondrial biogenesis and ROS removal [152].

In a recent study of the Nie group, it has been demonstrated that the deficiency of miR-133a in mice leads to low levels of PGC-1α and nuclear respiratory factor-1(Nrf1), and lower mitochondrial mass and exercise tolerance [83]. Since this phenotype is similar to the sarcopenia phenotype, the authors speculate that miR-133a might have a significant role in maintaining skeletal muscle mitochondrial functionality. Other miRNAs, such as miR-340-5p and miR-206, have also been shown to regulate ROS generation in skeletal muscle via Nrf2, which is a key factor in regulating redox homeostasis, although the molecular mechanisms involved in its effect in the onset of sarcopenia are still unknown [153,154].

Since it has been widely demonstrated that nutrients may influence the expression of endogenous miRNAs involved in different cellular processes, the manipulation of miRNAs profiles through dietary modifications and supplements can be proposed as a potential future therapeutic intervention or prevention strategy against sarcopenia.

## 5. High Fructose Diet Modulation of miRNAs Expression in Sarcopenia

Among the nutritional factors that have been reported to play a crucial role to increase inflammation [155], mitochondrial dysfunction and ROS production in skeletal muscle is fructose [156,157,158,159]. Fructose is one of the major constituents of the modern diet, since it is highly expressed in fruits and vegetables [160] and it is also used as a sweetener for food and drinks, and as an excipient in pharmaceutical preparations, syrups, and solutions [161].

Although low doses of fructose have beneficial effects on glycemic control without increasing cardiometabolic risk [162] and blood pressure [163], several studies demonstrated that a high level of fructose can stimulate ROS production in the mitochondria in a variety of tissues including kidney, liver, small intestine and skeletal muscle [164,165,166,167,168,169]. Fructose can exert these effects in different ways, including increased blood uric acid (UA) concentration [170,171], with a consequent upregulation of TGF-β1 expression and NOX4 activation [172] and through the induction of de novo lipogenesis [173,174,175,176].

In recent years, a gradual increment in blood UA concentration has been demonstrated, especially in people of Western countries, where the increased consumption of fructose has been revealed [177,178]. Excessive fructose consumption has also been associated with hepatic steatosis, cellular stress, and inflammation [179]. This is responsible for the release by the liver of lipids, methyglyoxal, UA, and hepatokines leading to alterations in the communication between the liver and the gut, muscles, and adipose tissue. Fructose and muscle/liver axis has been reported in several studies that showed how a high-fructose diet is associated with modifications in muscle function [180] in humans [181] and rodents [182]. In particular, mechanisms involved in diet-induced sarcopenia may be (i) a decrease in the mechanistic target of rapamycine complex (mTORC) 1 activity and thereafter in protein synthesis; and (ii) inflammation. Moreover, recent studies in fructose-fed rats have shown an association between nonalcoholic-fatty liver disease and sarcopenia [183]. This is a key factor involved in disease progression to NASH (nonalcoholic steatohepatitis), as the muscle heavily contributes to energy homeostasis [184].

In the inter-organs crosstalk caused by excessive fructose intake, it is absorbed primarily in the gut, and then metabolized in the liver, where it stimulates UA production [185,186]. The increased levels of intracellular UA are followed by an acute rise in circulating levels of UA, which is likely due to its release from the liver [170,171]. Besides, fructose may stimulate UA synthesis from amino acid precursors such as glycine [187], and it has been reported that long-term fructose administration suppresses the renal excretion of UA, resulting in elevated serum UA levels [188]. Interestingly, Kaneko and colleagues found that a single administration of fructose affects the excretion of UA to the intestinal lumen, inducing the reactive oxygen species (ROS)-derived production of dinucleotide phosphate (NADPH) oxidase activation [189].

Besides, experimental evidence shows that fructose-dependent UA production stimulates the upregulation of TGFβ-1, leading to NOX4 activation and ROS generation in mitochondria in skeletal muscle [172,190] (Figure 3).

As mentioned above, fructose consumption increases de novo lipogenesis in the liver, that is accompanied by an increased release of lipids in the bloodstream, which are then uptaken by different tissues, such as skeletal muscle [173,174,175,176]. In skeletal muscle, intracellular lipids accumulation increases the production of ROS and reactive nitrogen species [191,192,193]. Furthermore, the excessive production of lipoproteins induces an inflammatory response and, consequently, an elevation in circulating fatty acids and inflammatory cytokines, that may cause insulin resistance in peripheral tissues, leading to whole-body insulin resistance [194,195,196] (Figure 3). In a recent paper, Tamrakar group reported that, in myogenic cells, the fructose-dependent ROS production results in the activation of the stress/inflammation markers c-Jun N terminal kinase (JNK) and extracellular signal-regulated kinase 1/2 (ERK1/2), and the degradation of inhibitor of NFκB (IκBα), leading to impaired insulin signaling and attenuated glucose utilization in skeletal muscle cells [156].

Although the role of fructose in causing energy alterations and metabolic disorders has been well documented, the molecular mechanisms that regulate these effects have not yet been elucidated. In recent years, a growing interest has been directed to the role of miRNAs, since they are known to be dysregulated in several metabolic disorders and sarcopenia, and can be controlled by dietary factors [197,198]. A study of Su group demonstrated that a set of miRNAs are altered by high fructose diet; among them, miRNA-101a, miRNA-30a, and miRNA-582 have been reported to be involved in other cellular processes than energy metabolic signaling [199]. For example, fructose induces the expression of miRNA-101 involved in skeletal muscle cell proliferation and differentiation [200], and of miR-30a, which belongs to a miRNAs family, that promotes skeletal muscle differentiation. Both miRNAs are down-regulated in in vivo models of muscle injury and muscle disuse atrophy [201].

Besides, a high fructose diet may regulate a set of miRNAs involved in the hepatic insulin signaling. Among them, miR-128a can regulate insulin receptor substrate 1 (IRS1), ultimately affecting glucose and lipid metabolism [202]. Interestingly, the modulation of IRS1 by miR-128a has been reported in skeletal muscle, where it regulates myoblast proliferation and myotube hypertrophy and provides a novel mechanism, through which IRS1-dependent insulin signaling is regulated in skeletal muscle [199].

In summary, these data demonstrate that a high fructose diet can induce metabolic dysfunctions and modulate several processes, including oxidative stress and inflammation, that are also characteristic of sarcopenic muscles. In this contest, a crucial role is played by miRNAs, that can be altered by a high fructose diet, providing novel insights to counteract the physio-pathological effect of aging in different tissues (Figure 3).

## 6. Circulating miRNAs

Circulating miRNAs (c-miRNAs) represent a category of non-coding RNAs detectable in different bio-fluids, such as saliva, breast milk, urine, plasma, and serum [203,204]. Several mechanisms have been demonstrated for c-miRNAs packing and secretion, avoiding their degradation by serum ribonuclease. These include exosomes [205,206], high-density lipoprotein [207], RNA-binding proteins [208,209], and apoptotic bodies [210]. C-miRNAs can actively participate in cell-cell communication in different organs and tissues and, since it has been reported that their expression can be altered in pathological conditions and ageing, they have been suggested as potential biomarkers for the diagnosis and treatment of several diseases [211,212,213,214,215,216].

In particular, several papers in the last years have demonstrated a differential expression of c-miRNAs in sarcopenic compared to non-sarcopenic patients [58,217]. Besides, a correlation between nutrition, c-miRNAs, and sarcopenia has been recently revealed [58,218], highlighting a new role for the c-miRNAs as potential noninvasive biomarkers for the diagnosis of sarcopenia and the involvement of nutrition in this contest.

A number of studies have shown that miRNAs can be derived not exclusively from endogenous synthesis, but might also be obtained from dietary sources such as plants and animal origin food. These miRNAs are known as xeno-miRNAs [219,220,221,222]. Recently, it has been revealed by Zhang et al. that an exogenous plant-derived miRNA, miR168a, is one of the most highly enriched exogenous plant miRNA found in the serum of Chinese subjects. By using both in vivo and in vitro experimental models, they demonstrated that miR168a, packaged into microvesicles (MVs), can pass through the mouse gastrointestinal tract and might be released in the circulatory system, decreasing the plasma level of low-density lipoprotein [219]. In the same years, further research has shown that about 100 miRNAs are present in bovine milk that are resistant and stable to both industrial procedures and harsh conditions (low pH and RNase treatment). These miRNAs, encapsulated into MVs, can be diffused among animal species by dietary means, and are able to regulate a variety of metabolic pathways in humans and rats [223,224].

These data suggest that xeno-miRNAs may contribute to the circulating miRNAs population and, thanks to their effect in the modulation of target gene expression and the maintenance of tissue homeostasis, can represent novel biomarkers of age-related muscle mass and functionality.

## 7. Conclusions

Since sarcopenia dramatically affects the quality of life of older adults, therapeutic strategies are needed to prevent and/or counteract the progressive age-related reduction in skeletal muscle mass and functionality. Accumulating evidence suggests that nutrients such as amino acids, vitamins, and antioxidants represent key tools to elicit anabolic signaling and protein turnover, favoring the maintenance of muscle function. Dietary compounds have been shown to influence miRNAs levels in skeletal muscle and, given the importance of miRNAs as crucial regulators of skeletal muscle mass, composition and function, they may represent diagnostic or prognostic biomarkers of age-related muscle dysfunctions (Table 1).

## Figures and Tables

**Figure 1 antioxidants-09-00951-f001:**
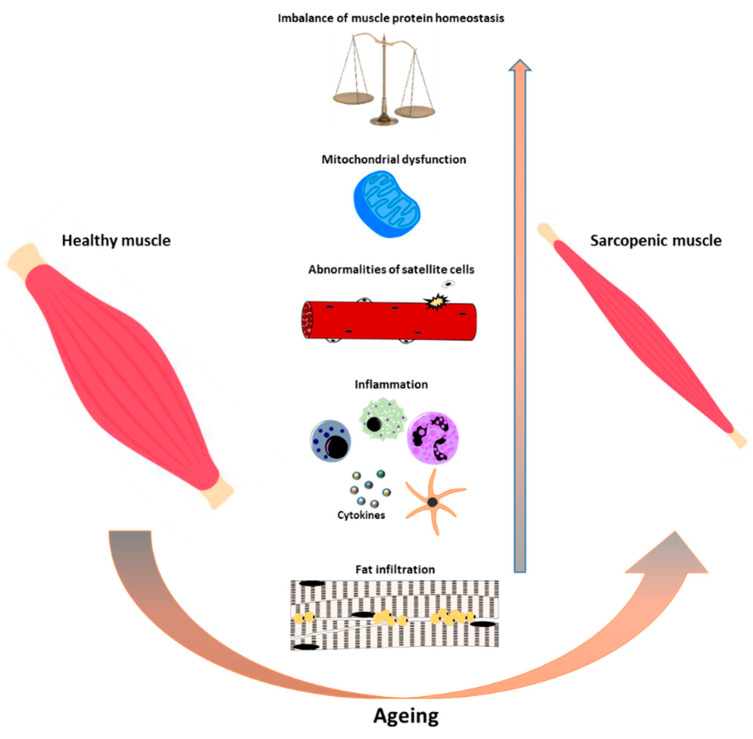
Schematic representation of cellular processes involved in the onset of sarcopenia during ageing. During ageing multifactorial events such as protein synthesis/degradation imbalance, satellite cell number/activity impairment, chronic inflammation, mitochondrial dysfunction, and fat infiltration increase contributing to the onset of sarcopenia.

**Figure 2 antioxidants-09-00951-f002:**
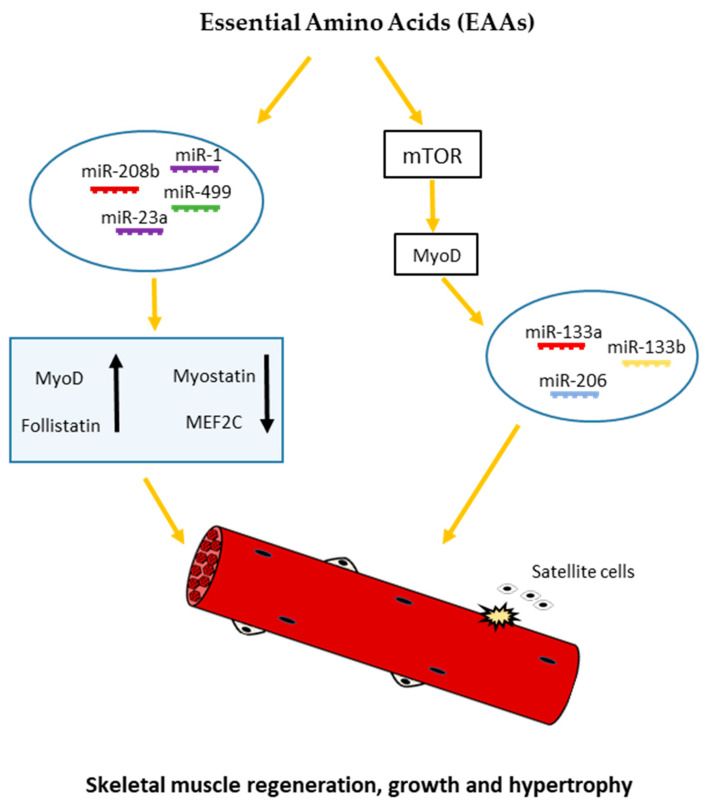
Schematic representation of nutrient-dependent-miRNA signaling in skeletal myogenesis. Nutrients such as essential amino acids (EAAs) may affect the expression of miR-133a/b and miR-206 through mTOR-dependent regulation of MyoD mRNA levels. On the other hand, EAAs may elicit robust increases in miR-1, miR-23a, miR-208b, and miR-499 expression, with an accompanying increase in MyoD and follistatin mRNA and decrease in myostatin and MEF2C mRNA expression, regulating skeletal muscle growth and differentiation.

**Figure 3 antioxidants-09-00951-f003:**
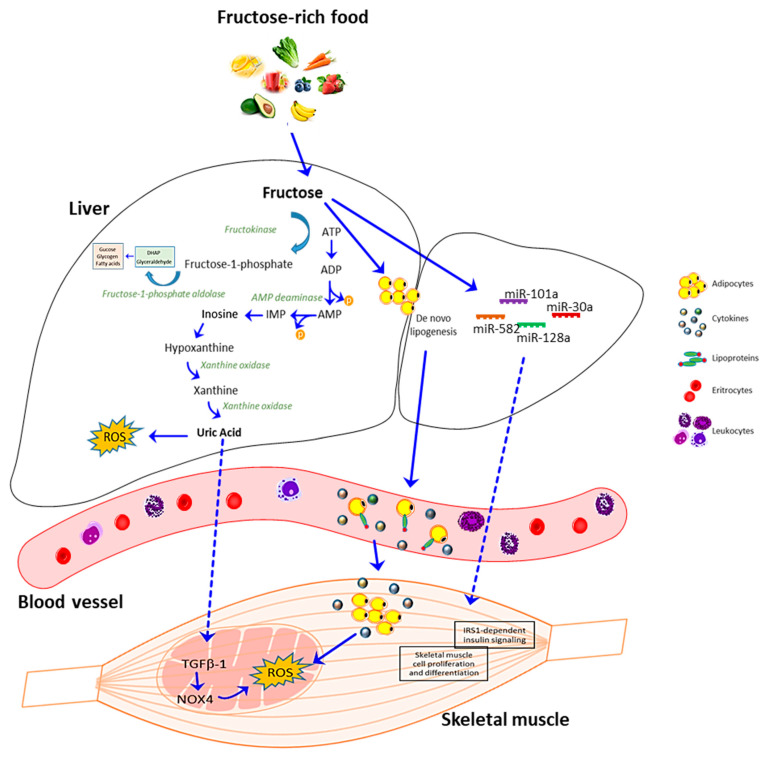
Schematic representation of fructose metabolism and uric acid effect on skeletal muscle. In the liver, fructose is phosphorylated into fructose 1-phosphate by fructokinase in a reaction that decreases the levels of intracellular phosphate and ATP. Subsequently, the enzyme fructose-1-phosphate aldolase gives rise to dihydroxyacetone phosphate (DHAP) and glyceraldehyde. When fructose 1-phosphate accumulates, intracellular phosphate decreases, stimulating AMP deaminase, which catalyzes the degradation of AMP to inosine monophosphate (IMP). IMP is metabolized to inosine, which is further degraded to xanthine and hypoxanthine by xanthine oxidase, ultimately generating uric acid (UA). UA can induce ROS production in the liver and other tissues, such as skeletal muscle via TGFβ-1-NOX4 signaling. Alternatively, fructose can stimulate de novo lipogenesis in the liver with increased release of lipids and lipoproteins in the bloodstream that are then uptaken by different tissues including skeletal muscle, with consequent cytokines and ROS production. Besides, high levels of fructose can modulate the expression of miRNAs that may affect skeletal muscle cell proliferation, differentiation, and insulin signaling.

**Table 1 antioxidants-09-00951-t001:** Dietary compounds that have been shown to influence miRNAs levels in skeletal muscle.

Nutraceuticals	$↑\begin{array}{c} \text{Positive}\\ \text{MyomiRs}\\ \text{Modulation} \end{array}$	$↓\begin{array}{c} \text{Negative}\\ \text{MyomiRs}\\ \text{Modulation} \end{array}$	Final Effect	References
EAAs	miR-1, miR-23a, miR-208b, miR-499, miR-27a		Skeletal muscle regeneration, proliferation and differentiation	[52,55]
Serum iron	miR-133b		Skeletal muscle differentiation	[58]
Albumin	miR-133b, miR-206		Skeletal muscle regeneration and differentiation	[58]
Ferritin		miR-133b, miR-206	Downregulation of skeletal muscle regeneration and differentiation	[58]
Insulin		miR-1, miR-133a, miR-206, miR-29a, miR-29c	Downregulation of skeletal muscle regeneration, differentiation and insulin resistance	[225]
Resveratrol	miR-21, miR-27b	miR-133b, miR-20b, miR-149	Modulation of skeletal muscle differentiation	[198,226,227]
Palmitic Acid	miR-29a		Insulin resistance and diabetes	[228]
Vitamin D	miR-26a		Skeletal muscle regeneration and differentiation	[229]
Fructose	miR-101, miR-30a	miR-128a	Skeletal muscle proliferation and differentiation, insulin signaling, insulin resistance	[199]

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
