# Peer review of "Nutrition and microRNAs: Novel Insights to Fight Sarcopenia"

_antioxidants, 2020, doi:10.3390/antiox9100951_

Round 1

Reviewer 1 Report

This is an interesting review that has summarized the role of nutrition-dependent regulation of microRNA changes in sarcopenia. Authors have done a good job of dividing the sections and focusing on the important area of inflammageing and mitochondrial dysfunction. The comment below will help to improve the quality of the review:

Major comments:

  • Authors have listed many factors that contribute to inflammageing and also mitochondrial dysfunction. One nutritional factor that can contribute to both of these factors and link the dots together in humans is uric acid. It would be very helpful that authors have a separate section and give an example of a translational nutritional factor that can not only contribute to systemic inflammation and mitochondrial dysfunction/ROS imbalance but also have been linked to sarcopenia. As mentioned above the potential role of Xanthine Oxidoreductase and Uric acid to this axis could be novel. Authors can use the current literature regarding the role of high fructose consumption in hyperuricemia and the potential link to sarcopenia.

Minor comments:

  • It would be very helpful if authors also add a third figure where they include nutritional factors (e.g. High fructose/Western diet?) and link them the potential causal link (e.g. hyperuricemia) which can link mitochondrial dysfunction, ROS imbalance and systemic inflammation to potential changes in microRNA in sarcopenia.

Author Response

Reviewer #1 major comments:

Authors have listed many factors that contribute to inflammageing and also mitochondrial dysfunction. One nutritional factor that can contribute to both of these factors and link the dots together in humans is uric acid. It would be very helpful that authors have a separate section and give an example of a translational nutritional factor that can not only contribute to systemic inflammation and mitochondrial dysfunction/ROS imbalance but also have been linked to sarcopenia. As mentioned above the potential role of Xanthine Oxidoreductase and Uric acid to this axis could be novel. Authors can use the current literature regarding the role of high fructose consumption in hyperuricemia and the potential link to sarcopenia.

Authors’ Response:

According to the major comments of the Reviewer 1, we introduced a new section entitled: “High fructose diet modulation of miRNAs expression in sarcopenia” in which we analyzed the role of high fructose diet in inducing metabolic dysfunctions and in the modulation of miRNAs involved in several processes including oxidative stress and inflammation, characteristic of sarcopenic muscles. In the text, this part is highlighted in yellow (pages 8-10).

Reviewer 1 minor comments:

It would be very helpful if authors also add a third figure where they include nutritional factors (e.g. High fructose/Western diet?) and link them the potential causal link (e.g. hyperuricemia) which can link mitochondrial dysfunction, ROS imbalance, and systemic inflammation to potential changes in microRNA in sarcopenia.

Authors’ Response:

We also introduced a new Figure (Fig.3) in which we schematized a potential link between nutritional factors such as fructose, miRNAs, and several processes known to be dysregulated in sarcopenia. This Figure is placed after the “High fructose diet modulation of miRNAs expression in sarcopenia” section (pages 9 and 10).

Reviewer 2 Report

The reviewed manuscript is interesting. The authors described the most relevant molecular mechanisms related  to the pathophysiological effect of sarcopenia. They focus on the role of nutrition as a possible way to counteract the loss of muscle mass and function associated with ageing with a special  attention to nutrient-dependent miRNAs regulation.

I have just one suggestion. It would be nice if the authors would provide a table with name of nutraceuticals (like flavonoids, carotenoids, etc.) that can modulate miRNA-1, miRNA-133a, miRNA-55 133b, miRNA-206, miRNA-208b, miRNA-486 and miRNA-499 . It will be useful for many readers.

Author Response

Reviewer #2 comments:

I have just one suggestion. It would be nice if the authors would provide a table with the name of nutraceuticals (like flavonoids, carotenoids, etc.) that can modulate miRNA-1, miRNA-133a, miRNA-55 133b, miRNA-206, miRNA-208b, miRNA-486, and miRNA-499. It will be useful for many readers.

Authors’ Response:

As suggested by this Reviewer, we introduced a table in which we reported a list of specific nutrients and their effect in the modulation of skeletal muscle processes through the myomiRNAs regulation. The table is placed after the conclusion section (page 11).

Reviewer 3 Report

This manuscript presents many kinds of molecular mechanism and physiological condition related to nutrition and microRNA to be able to control sarcopenia

In the last 5 years, too many review paper for the subject of ‘nutrition and sarcopenia’ have been published, compared to research papers. Moreover, as the authors mentioned in line 196, there are only a few studies about the interconnection between ‘miRNAs, nutrition and sarcopenia’. This means that reviewing for those subjects is too early to get valuable information and ideas.

Specific comments:

  1. Figure 1 is a general scheme that shows well-known cellular events in sarcopenia, which is dispensable in context, because a similar scheme appears in a number of review papers on sarcopenia. Authors should add the knowledge they gathered relating to nutrient-dependent microRNAs regulation as they mentioned.
  2. Figure 2 is totally incorrect. For example, myostatin and follistatin are circulating factors, which should be drawn extracellularly. Either amino acids of follistatin does not directly affect the expression of miRs. mTOR does not directly activate myoD or follistatin, but activates protein synthesis. Decreased expression of myostatin does not inhibit muscle inhibit muscle differentiation. Authors should make figures on the basis of direct evidences. Too much speculation should mislead readers.
  3. Even though the title is ‘Nutrition and microRNA’ and subtitle is ‘nutrition-dependent microRNA ~’, most of the content is just about the signaling pathway in skeletal muscle.
  • Line 94 -; The role of miRNA in muscle is described, but there is no content about nutrition dependency.
  • Line 106 -; Cited references about nutrition and miRNA are just reports about phenomena without the study on causality to demonstrate the relationship between nutrition and miRNA.
  • Line 136 - 153; Content is just about inflammageing, not nutrient and miR.
  • Line 154 - 164; Content is too simple to explain protein degradation. Other signaling pathways are also related to protein degradation, such as myotatin, TGFb signaling and so on, as well as NF-kB signaling,
  • Line 165 - 187; It is just about miRNA regulation in skeletal muscle, without nutrition dependency.
  • Page 6 - 7; ‘nutrition-dependent microRNA regulation of mitochondria dysfunction’ part also does not include ‘nutrition-dependent’ microRNA.
  • Page 8; Circulation miRNAs part of is not related to topic of this manuscript.

  1. Authors conclusion that nutrients affect microRNAs, inflammation, and finally sarcopenia in skeletal muscle is not convincing
  • Line 188 - 191; References cited have shown that nutrition affects the levels of several microRNAs, but does not show whether the microRNAs affect inflammation in skeletal muscle. So it is hard to say that malnutrition causes microRNAs expression which in turn results in inflammation-mediated sarcopenia
  • Line 106 – 107; Reference 51 shows that hyper glucose induces miR-21, which increase TORC1 activation mediated renal cell hypertrophy, a pathological feature of diabetic kidney disease. Therefore authors here misinterpreted several points here, considering skeletal muscle physiology and pathology. First, microRNA expression is tissue specific. Second, TORC1 activation is well known to induce muscle hypertrophy by elevation of protein synthesis but not by inhibition of inflammation. Moreover, miR-21 level has no correlation with inflammation within skeletal muscle. So, author’s conclusion is totally different from scientific evidences.
  • Reference 52 showed that miRNA (miR-499, -208b, -23a, -1, and pri-miR-206) and growth related genes (MyoD and FSTL1) expression increased 3h after 10g EAA ingestion in vastus lateralis muscles. Even there is no correlation data whether theses microRNA regulates those growth related genes. So, here authors cannot say any link between nutrition and inflammation in sarcopenia.
  • Reference 53 showed leucine induced microRNA-27 expression and C2C12 proliferation. However, the cause-and-effect was rather unlikely.
  • Line 185 – 187; Authors claim that miR-181 was downregulated in old muscle resulting in increased expression of inflammatory cytokines (Reference 84). In the reference 84, however, miR-181a serves as an inducer of inflammatory cytokines in blood immune cells. No data with skeletal muscles.
  • Line 188 – 195; Correlation of nutrition and miRNA are presented, but it is just phenomena without mechanical link.
  • Line 300 ~ 303; References are not enough to conclude like this paragraph.

Conculsion:

  • Although topic is very interesting, it is hard to gather the right references that fit in this subject, to provide helpful information for the link between nutrition and microRNA involved in sarcopenia pathologies yet.
  • Main focus of this manuscript is not suitable for the journal scope named ‘Antioxidants’.

Author Response

From the Editor

Although we did not trust on the report of reviewer 3 which was rather negative, I am giving you his specific comments and hope that they help you to further improve your manuscript:

Authors’ Response:

Regarding the comments of Reviewer #3, we have considered some of his suggestions. In particular:

  • we modified Figure 2 simplifying the schematization of the processes involved in the nutrient-dependent-miRNA signaling in skeletal muscle myogenesis;
  • by adding the section related to high fructose diet modulation of miRNAs expression in sarcopenia we better explained the role of nutrition-dependent-miRNA regulation of mitochondrial dysfunctions.